# α-Bisabolol Attenuates Doxorubicin Induced Renal Toxicity by Modulating NF-κB/MAPK Signaling and Caspase-Dependent Apoptosis in Rats

**DOI:** 10.3390/ijms231810528

**Published:** 2022-09-10

**Authors:** Seenipandi Arunachalam, M. F. Nagoor Meeran, Sheikh Azimullah, Niraj Kumar Jha, Dhanya Saraswathiamma, Sandeep Subramanya, Alia Albawardi, Shreesh Ojha

**Affiliations:** 1Department of Pharmacology and Therapeutics, College of Medicine and Health Sciences, United Arab Emirates University, Al Ain P.O. Box 15551, United Arab Emirates; 2Department of Biotechnology, School of Engineering and Technology (SET), Sharda University, Greater Noida 201310, UP, India; 3Department of Pathology, College of Medicine and Health Sciences, United Arab Emirates University, Al Ain P.O. Box 15551, United Arab Emirates; 4Department of Physiology, College of Medicine and Health Sciences, United Arab Emirates University, Al Ain P.O. Box 15551, United Arab Emirates; 5Zayed Bin Sultan Center for Health Sciences, United Arab Emirates University, Al Ain P.O. Box 15551, United Arab Emirates

**Keywords:** α-Bisabolol, doxorubicin, nephrotoxicity, NF-κB/MAPK signaling, apoptosis, phytochemicals, renal toxicity

## Abstract

Doxorubicin (DOX) is a well-known and effective antineoplastic agent of the anthracycline family. But, multiple organ toxicities compromise its invaluable therapeutic usage. Among many toxicity types, nephrotoxicity is one of the major concerns. In recent years many approaches, including bioactive agents of natural origin, have been explored to provide protective effects against chemotherapy-related complications. α-Bisabolol is a naturally occurring monocyclic sesquiterpene alcohol identified in the essential oils of various aromatic plants and possesses a wide range of pharmacological properties such as antioxidant, anti-inflammatory, analgesic, cardioprotective, antibiotic, anti-irritant, and anticancer activities. The present study aimed to evaluate the effects of α-Bisabolol on DOX-induced nephrotoxicity in Wistar male albino rats. Nephrotoxicity was induced in rats by injecting a single dose of DOX (12.5 mg/kg, i.p.), and the test compound, α-Bisabolol (25 mg/kg) was administered intraperitoneally along with DOX as a co-treatment daily for 5 days. DOX-injected rats showed reduction in body weight along with a concomitant fall in antioxidants and increased lipid peroxidation in the kidney. DOX-injection also increased levels/expressions of proinflammatory cytokines namely tumor necrosis factor-α (TNF-α), interleukin-6 (IL-6), and interleukin-1β (IL-1β) and inflammatory mediators like inducible nitric oxide synthase (iNOS) and cyclooxygenase-2 (COX-2) and activated nuclear factor kappa-B (NF-κB)/mitogen-activated protein kinases (MAPK) signaling in the kidney tissues. DOX also triggered apoptotic cell death, evidenced by the increased expression of pro-apoptotic markers like BCL2-Associated X Protein (Bax), cleaved caspase-3, caspase- 9, and cytochrome-C) and a decrease in the expressions of anti-apoptotic markers namely B-cell lymphoma 2 (Bcl2) and B-cell lymphoma-extra large (Bcl-xL) in the kidney. These biochemical alterations were additionally supported by light microscopic findings, which revealed structural alterations in the kidney. However, treatment with α-Bisabolol prevented body weight loss, restored antioxidants, mitigated lipid peroxidation, and inhibited the rise in proinflammatory cytokines, as well as favorably modulated the expressions of NF-κB/MAPK signaling and apoptosis markers in DOX-induced nephrotoxicity. Based on the results observed, it can be concluded that α-Bisabolol has potential to attenuate DOX-induced nephrotoxicity by inhibiting oxidative stress and inflammation mediated activation of NF-κB/MAPK signaling alongwith intrinsic pathway of apoptosis in rats. The study findings are suggestive of protective potential of α-Bisabolol in DOX associated nephrotoxicity and this could be potentially useful in minimizing the adverse effects of DOX and may be a potential agent or adjuvant for renal protection.

## 1. Introduction

Doxorubicin (DOX), an anthracycline anticancer drug, has remarkable popularity in its clinical usage for managing various solid tumors, including ovary, breast, lung, cervix, uterine, as well as hematological malignancies. But its clinical usage has been constrained owing to its significant adverse effects, including nephrotoxicity [1,2]. The primary role of DOX is to induce cytotoxicity via DNA intercalation and topoisomerase II inhibition in rapidly growing malignant cells [3]. Although the specific mechanism behind DOX-induced renal injury is still unknown, available evidences reported that DOX-induced nephrotoxicity is ascribed to the accumulation of free radicals, ultimately inducing the membrane lipid peroxidation and protein oxidation [4]. Moreover, various recent studies have shown that inflammation and apoptosis also play a major role in DOX-induced renal injury [1].

DOX enhances oxidative stress by inducing the overproduction of free radicals, which causes renal tubular degeneration leading to renal dysfunction [5]. Inflammation is a counter reaction by the immune system against many factors like injured cells, harmful chemicals, and pathogens [6]. Under inflammatory conditions, macrophages altered the presence of inflammatory cytokines and mediators [7]. Several studies reported the relationship between DOX and nuclear factor kappa B (NF-κB), a transcriptional factor that regulates genes that encodes apoptosis and inflammatory cytokines [8]. Also, mitogen-activated protein kinase (MAPK) signaling, including p38 MAPK, plays a vital role in the modulation of inflammatory cytokines [9]. In addition, to modulate inflammatory cascades, P38 MAPK also possesses the ability to control cell cycle and apoptosis. DOX has been reported to trigger P38 MAPK activation, thus promoting the incidence of apoptosis [10].

It has been reported that nitric oxide synthase (NOS) activation facilitated apoptosis are additional sources that may involve lethal consequences accompanying DOX- chemotherapy [5]. Involvement of DOX in nitric oxide (NO) metabolism through direct or indirect stimulation of nitric oxide (NO) production, which triggers the production of free radicals. DOX reported totrigger free radical production and the release of NO directly contribute to DOX-induced nephrotoxicity [11]. Also, DOX might encourage renal toxicity through its deleterious effects on the kidney by affecting the permeability of the glomerulus and induces tubular degeneration [12]. DOX causes toxicity to the heart and liver, possibly alters blood circulation to the kidney, and changes xenobiotic reclamation, subsequently accomplishing nephropathy [5].

In recent years, sesquiterpenes, a class of phytochemicals abundantly present in aromatic plants, have been recognized as the most active constituents, showing diverse biological activities including chemopreventive, and anticancer as well as organ-protective owing to antioxidant and anti-inflammatory properties [13]. α-Bisabolol, also known as levomenol, is one of the major unsaturated monocyclic sesquiterpene alcohol, mainly present in Chamomile (*Chamomilla recutita* L.) [14], Candeia wood (*Eremanthus erythropappus*) [15], *Plinia cerrocampanensi* [16] and Salvia (*Salvia runcinata*) [13]. It is a clean, colorless liquid with a nutty fruit aroma, and a low-density compound exists in two configurations, ά and β. Most of its biological properties are attributed to its ά-isomer form. It is capable of undergoing easy oxidation due to its high lipophilic nature and acquired the ability to produce two bisabolol oxides (A and B). It also possesses a wide range of pharmacological properties like antioxidant, anti-inflammatory, anti-microbial, cardioprotective, antinociceptive, and neuroprotective properties [17].

α-Bisabolol is a vital ingredient in cosmetic and dermatological formulations and is found to be safe and non-toxic when administered orally to the rodent species (LD_50_ 13–14 g/kg body weight) [17]. It has been approved by regulatory agencies for use in cosmetics and food products as an additive and flavoring agent considering its dietary safety within normal limits. It has been shown to possess potent anticancer activity, as demonstrated in experimental cancer models [17]. It has been suggested for its use as a possible combination of conventional therapies in breast cancer and surgical tumor removal (adjuvant therapy). It has also been shown cardioprotective in isoproterenol-induced myocardial injury [18] and nephroprotective in cisplatin-induced renal injury [19]. A recent review has highlighted its sources and potential health benefits, and therapeutic properties [20]. Despite its potential role in cancer and cardiovascular diseases, there are no data available on the role of α-Bisabolol against DOX-induced nephrotoxicity in rats. In the present study, the role of α-Bisabolol was investigated in DOX-induced renal injury in rats, and underlying molecular mechanisms were studied to determine its effect on oxidative stress, inflammation, apoptosis, and NF-κB/MAPK signaling pathways. 

## 2. Results

### 2.1. α-Bisabolol Rescued Weight Loss and Renal Function in DOX-Induced Renal Injury

DOX-injections caused a severe decrease in body weight, while α-Bisabolol reinstated the observed body weight loss in DOX-injected rats. Additionally, DOX-injections triggered a considerable (*p* < 0.05) increase in the levels of serum creatinine, while α-Bisabolol treatment showed a considerable (*p* < 0.05) decrease in the levels of this renal injury marker in DOX-injected rats (Figure 1A,B).

### 2.2. α-Bisabolol Inhibited Oxidative Stress in DOX-Induced Renal Injury

DOX-injected rats showed a significant (*p* < 0.05) increase in the levels of renal MDA with a considerable (*p* < 0.05) decrease in the activities/concentration of renal SOD, catalase, and GSH. α- Bisabolol treatment considerably (*p* < 0.05) decreased the renal MDA content and marked a subtle increase in the activities/levels of renal SOD, catalase, and GSH in DOX-injected rats compared to DOX-alone treated rats (Figure 1C–F).

### 2.3. α-Bisabolol Inhibits the Levels/Expressions of Pro-Inflammatory Cytokines in the Kidneys

Renal tissue levels/expressions of TNF-α, IL-6, and IL-1β were significantly (*p* < 0.05) increased compared to normal control rats. α-Bisabolol treatment significantly (*p* < 0.05) reversed the DOX-triggered rise in the levels/expressions of renal proinflammatory cytokines compared to DOX-alone injected rats (Figure 2 and Figure 3).

### 2.4. α-Bisabolol Protects the Renal Architecture in DOX-Injected Rats

Renal sections of normal and α-Bisabolol alone treated rats showed no considerable changes in the renal architecture. However, histological evaluation of renal sections in DOX-treated rats showed thinning and loss of apical cytoplasm, necrosis of individual tubular epithelial cells, loss of nuclei, adhesion, and denudation of tubular basement membrane, whereas α-Bisabolol treatment reinstates the near normal renal architecture in rats which revealed its membrane stabilizing property (Figure 4).

### 2.5. α-Bisabolol Attenuates the Expressions of Inflammatory Mediators and Downregulates NF-κB/MAPK Signaling Pathway

The expressions of renal iNOS, COX-2, p-NF-κB, p-IκB, and p-P^38^ in DOX-alone treated rats were considerably (*p* < 0.05) increased compared to normal rats. α-Bisabolol treatment considerably (*p* < 0.05) reduced the increased renal expressions of iNOS, COX-2, p-NF-κB, p-IκB, and p-P^38^ compared to DOX-alone treated rats (Figure 5A,B).

### 2.6. α-Bisabolol Attenuates Caspase-Dependent Renal Apoptosis

DOX-injected rats showed a significant (*p* < 0.05) increase in the renal protein expressions of Bax, cleaved caspase-3, cleaved caspase-9, and cytochrome-C with a considerable (*p* < 0.05) decrease in the expressions of Bcl2 and Bcl-xL compared to normal control rats. Meanwhile, α-Bisabolol treatment to DOX-injected rats significantly (*p* < 0.05) downregulates the renal protein expressions of Bax, cleaved caspase-3, cleaved caspase-9, and cytochrome-C with significant (*p* < 0.05) decrease in the expression of renal Bcl2 and Bcl-xL was observed. The results have revealed that α-Bisabolol protects the kidney by modulating caspase dependent apoptosis in DOX-injected rats (Figure 6A,B).

## 3. Discussion

The present study findings demonstrate the nephroprotective property of α-Bisabolol against DOX-induced renal injury attributing to its potent antioxidant, anti-inflammatory, anti-apoptotic, and membrane stabilizing properties. Nephrotoxicity is one of the major side effects after DOX treatment, mainly due to renal oxidative stress in cancer patients [5]. Also, DOX possess the ability to accumulate in both kidneys and triggers direct catastrophic effect [21]. Involvement of oxidative stress, inflammation, and apoptosis are the major mechanisms behind DOX-mediated renal toxicity [22]. Also, the semiquinone form of DOX and its involvement in creating superoxide radicals is another mechanism contributing to its toxic effect on kidneys [23]. To the best of our knowledge, this is the first scientific evidence reported on the effects of α-Bisabolol against DOX-induced nephrotoxicity in rats.

The reduction in body weight of DOX-injected rats is accompanied by anorexia (loss of appetite), progressive exhaustion with declined physical activity [24]. Also, DOX-induced renal toxicity was identified by increased serum creatinine levels, a highly sensitive sign of kidney damage concerned with the diagnosis of renal toxicity [25]. Thus, free radical-mediated oxidative damage plays a major role in DOX-induced renal toxicity. The near normal levels of serum creatinine along with rescued body weight in DOX-injected rats observed in our study revealed that this could be due to the potent antioxidant capacity of α-Bisabolol.

Much attention has been given to oxidative stress in our present study due to its vital role in DOX-induced kidney injury, as reported previously [26]. A report by [27] showed that the kidney is highly susceptible to oxidative stress due to the abundant concentration of non-heme iron catalytically involved in free radical production. Verily, the ring structure of DOX increases the enzymatic and non-enzymatic single-electron redox cycle release of ROS from molecular oxygen [28]. DOX-mediated free radicals deplete the antioxidant defenses (SOD, catalase, and GSH), which leads to the active mediation of lipid and protein oxidation [29]. α-Bisabolol treatment showed remarkable improvement of the antioxidant defenses with a consistent decrease in the lipid peroxidation products in response to DOX-insult. The well-documented free radical scavenging and antioxidant potential of α-Bisabolol should be correlated with the observed decrease in oxidative stress in kidney tissues.

NF-κB is a crucial transcriptional activator that regulates the expressions of various inflammation factors. It plays a major role in the pathology of the DOX-induced renal inflammation process [30,31]. ROS is the main initiator of NF-κB activation, which is responsible for the inflammatory cascades through the mediation of proinflammatory cytokines (TNF-α, IL-6, and IL-1β) and inflammatory mediators (iNOS and COX-2 expressions) [23]. The dominance of these proinflammatory mediators triggers NF-κB activation, and this positive feedback mechanism is speculated to amplify proinflammatory signals and exacerbate renal tissue injury [32]. NF-κB activation and phosphorylation by DOX are strongly linked with the involvement of IκK, which actively triggers the phosphorylation and degradation of IκB [33,34]. Under an environment of oxidative stress and inflammation, IκK and IκB activation might promote NF-κB phosphorylation, which results in irreversible inflammatory assault [35]. Along with NF-κB, MAPK family proteins are the key regulators of cellular differentiation, proliferation, cell death, and inflammatory mediators [36]. Upon inflammatory stimulation, MAPK P38 kinases are crucial in promoting inflammation in DOX-injected rats [37]. Interestingly, α-Bisabolol treatment showed very high-level resistance against DOX-induced renal damage by reducing the dominance of proinflammatory mediators along with NF-κB/MAPK signaling cascades revealing its potent anti-inflammatory effect. DOX-induced apoptosis is considered to be an initiator of a rise in the oxidative and inflammatory mediators, which activates pro-apoptotic signals. MAPK possesses the ability to motivate NF-κB and activates downstream genes, which further regulates the proinflammatory responses that could regulate the pathological condition. Ultimately, excess free radical production, upregulated cytokine production, and MAPK signaling cascade activates NF-κB transcription factors and apoptotic markers [8]. Activation of p38-MAPK subsequently induces cytochrome-C release from the mitochondria, triggering the intrinsic pathway of apoptosis [38,39]. It is very well known that the upregulation of pro-apoptotic marker BAX and the downregulation of anti-apoptotic marker Bcl2 along with caspase-3 activation are inevitable after cytochrome-C release from the mitochondria in DOX-invoked apoptosis. Our study is also witnessed the same changes in the renal protein expressions of apoptotic signaling markers in DOX-induced rats. It is evident from this study that the incidence of renal apoptosis is due to free radical mediated NF-κB activation and its influence over MAPK signaling to induce apoptosis directly linked with the development of DOX-induced nephrotoxicity. α-Bisabolol effectively defends the kidney from the rising apoptotic signals in DOX-induced renal toxicity clearly revealed its anti-apoptotic effect. The abovementioned biochemical and molecular evidences are found in line with the improvements noted in the histological study in which α-Bisabolol treatment effectively preserved the renal architecture in DOX-induced rats. This clearly revealed the membrane stabilizing and anti-apoptotic abilities of α-Bisabolol.

## 4. Materials and Methods

### 4.1. Drugs, Chemicals, and Antibodies

α-Bisabolol was purchased from Sigma-Aldrich (St. Louis, MO, USA). DOX was purchased from Sigma-Aldrich (St. Louis, MO, USA). Polyclonal rabbit and monoclonal mouse anti-inducible nitric oxide synthase (iNOS), cyclooxygenase-2 (COX-2) and anti-nuclear factor kappa B-p65 (NF-κB-p65), phospho-NF-κB-p65, Bcl2 associated X protein (Bax), B-cell lymphoma 2 (Bcl-2), active caspase-3 were purchased from Abcam (Cambridge, MA, USA), Cell signaling technology (Beverly, MA, USA) and Santa Cruz Biotechnology (Dallas, TX, USA). Secondary biotinylated and horseradish peroxidase-conjugated goat anti-rabbit and goat anti-mouse antibodies were obtained from Santa Cruz Biotechnology (Dallas, TX, USA).

### 4.2. Experimental Animals

Male albino Wistar rats (220–250 g) were acclimatized for two weeks before initiating the experimental protocol in the Animal Research Facility of the College of Medicine and Health Sciences, United Arab Emirates University. The animals were housed in a group of four under standard laboratory conditions of light and dark cycles with free access to commercially available rodent food and water *ad libitum*. Experimental procedures were conducted following the animal experimentation protocol approval by the Animal Ethics Committee of the United Arab Emirates University, Al Ain, Abu Dhabi, United Arab Emirates.

### 4.3. Experimental Design

The animals were randomly divided into four experimental groups: each containing fifteen rats. α-Bisabolol was diluted in scientific grade light olive oil (vehicle), and the solutions were freshly prepared just before administration. A single intraperitoneal injection of DOX (12.5 mg/kg body weight) was administered to the rats to induce nephrotoxicity. The dose of DOX is well standardized and established enough to induce nephrotoxicity as assessed by an increase in the levels of serum creatinine. Group-I: normal control rats; Group-II: rats treated with α-Bisabolol (25 mg/kg, intraperitoneally) daily for five days; Group-III: rats were intraperitoneally injected with a single dose of DOX (12.5 mg/kg) to induce nephrotoxicity; Group-IV: rats administered a single intraperitoneal dose of DOX (12.5 mg/kg) and α-Bisabolol (25 mg/kg, intraperitoneally) for five days. After the treatment duration (i.e., on the 6th day), all the rats were anesthetized with pentobarbital sodium (60 mg/kg, body weight) and then sacrificed by cervical decapitation. Serum samples were collected for the estimation of creatinine levels. The isolated kidneys were snap frozen in liquid nitrogen (LN2) and stored for the biochemical and immunoblotting experiments. The kidneys were also fixed in the 10% neutral buffered formalin for histological studies.

### 4.4. Biochemical Parameters

#### 4.4.1. *Estimation of Blood urea Nitrogen and Creatinine*

The serum creatinine levels were assayed using VetTest 8008 Chemistry Analyzer (UK).

#### 4.4.2. *Estimation of Lipid Peroxidation Products and Antioxidants*

The levels of renal malondialdehyde (MDA) were estimated following the manufacturer’s instructions provided in the commercially available detection kit (Northwest Life science, Vancouver, WA, USA). The activities of superoxide dismutase, catalase, and the concentration of glutathione were estimated according to the manufacturer’s instructions provided in the commercially available assay kits acquired from Sigma (St. Louis, MO, USA) and Cayman Chemical Company (Ann Arbor, MI, USA).

#### 4.4.3. *Estimation of Pro-Inflammatory and Anti-Inflammatory Cytokines*

The levels of tumor necrosis factor-α (TNF-α), interleukin-6 (IL-6), interleukin-1β(IL-1β), and interleukin-10 were measured by commercially available enzyme-linked immunosorbent assay (ELISA) kits obtained from BioSource International (Camarillo, CA, USA).

### 4.5. Western Blot Analysis

Protein extracts from kidney tissues were obtained by homogenizing kidney samples in an ice-cold radioimmuno precipitation assay buffer (RIPA) buffer supplemented with phosphatase and protease inhibitors (Sigma Aldrich, St. Louis, MO, USA), and the homogenates were centrifuged at 1648× *g* for 30 min at 4 °C. The supernatant was mixed with Laemmlli sample buffer (Bio-Rad, Hercules, CA, USA) and 2-mercaptoethanol (Sigma Aldrich, St. Louis, MO, USA). The samples containing equal amounts of protein were separated by gel electrophoresis and then transferred onto poly-vinylidene difluoride membranes (Amersham Hybond P 0.45, GE Health care Life Sciences, Munich, Germany). Membranes were incubated overnight at 4 °C with antibodies against inducible nitric oxide synthase (iNOS), Bax, Bcl2 (anti-rabbit; 1:1000 dilution; Sigma Aldrich, St. Louis, MO, USA), COX-2 (anti-rabbit; 1:500 dilution; Abcam, Cambridge, MA, USA), phospho NF-κB-p65 (anti-mouse; 1:1000 dilution; Santacruz, Dallas, TX, USA), active caspase-3 (anti-rabbit; 1:500 dilution; Abcam, Cambridge, MA, USA), cleaved caspase-9 (1:500 dilution; Cell signaling Technology, Beverly, MA, USA), GAPDH (anti-mouse; 1:2000 dilution; Merck Millipore, Burlington, MA, USA) was used as a loading control. The samples were then incubated with their corresponding secondary antibodies (Cell signaling Technology, Beverly, MA, USA) for 1h at room temperature, and the proteins were visualized by using an enhanced chemiluminescence kit (Thermo Fisher Scientific, Rockford, IL, USA). The signal intensity (densitometry) of the bands was quantified using Image J software.

### 4.6. Estimation of Protein Content in the Kidney

Protein contents in kidney homogenates were estimated using a Pierce™BCA protein assay kit (Thermo Fisher Scientific, Rockford, IL, USA).

### 4.7. Histopathological Evaluation

After fixation of kidney tissue in neutral buffered formalin (10% *w*/*v*) for a week, the tissue was gradually dehydrated in increasing concentrations of ethanol, cleared of alcohol residue in xylene, and finally embedded in paraffin. Sections of 5–10 μm were cut using a microtome (RM2125 RTS, Leica Biosystems, Nussloch, Germany) and stained with hematoxylin and eosin. Sections were mounted on slides and evaluated under a light microscope (BX41, Olympus, Tokyo, Japan) using a 20× objective lens.

### 4.8. Statistical Analysis

Statistical analysis was performed by one-way analysis of variance followed by Duncan’s Multiple Range Test (DMRT) using Statistical Package for the Social Science (SPSS) software v.25. Results are expressed as the mean ± standard error of the mean (S.E.M) for eight rats in each group. Differences between each group were considered significant at *p* < 0.05.

## 5. Limitations

In our study, we measured the activation of p38-MAPK but not p44/42 (ERK), which is considered the main MAPK protein, followed by NF-κB activation and apoptosis. But, we didn’t analyze the effect of α-Bisabolol with specific p38 inhibitors, which is a considerable limitation of the study.

## 6. Conclusions

Taken together, the α-Bisabolol treatment appears to protect against DOX-induced renal toxicity, and the mechanism underlying this nephroprotective effect is ascribed to its potent antioxidant, anti-inflammatory, anti-apoptotic, and membrane stabilizing properties. The protective properties have been evidenced by its ability to neutralize the free radicals produced during DOX metabolism and by favorable modulation of the NF-κB/MAPK signaling cascades and caspase-dependent apoptosis. Our study findings could encourage the use of α-Bisabolol as an agent or adjuvant for the prevention and treatment of renal injury associated with chemotherapeutic drugs in particular DOX. Furthermore, clinical studies are required in the future to confirm the findings of our study. This may be useful in preventing multiple organ dysfunction during chemotherapy and improving the lifestyle of cancer patients.

## Figures and Tables

**Figure 1 ijms-23-10528-f001:**
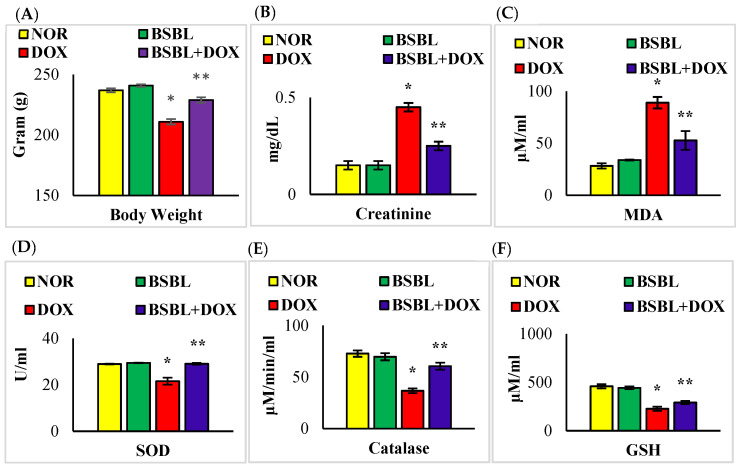
Effect of α-Bisabolol on body weight, renal injury, and oxidative stress markers in normal and DOX-injected rats. (**A**,**B**). Effect of α-Bisabolol on body weight and serum creatinine levels. DOX-injected rats showed a significant (*p* < 0.05) decrease in body weight along with a significant (*p* < 0.05) increase in the levels of creatinine in the serum compared to normal control rats, whereas α-Bisabolol treatment in DOX-treated rats significantly (*p* < 0.05) decreased the body weight loss and serum creatinine levels compared to DOX-alone treated rats (**C**–**F**). Effect of α-Bisabolol on the levels/activities/concentrations of MDA, SOD, catalase, and GSH in the kidney of normal and DOX-injected rats. DOX-injected rats showed a considerable (*p* < 0.05) decrease in the activities/levels of enzymatic and non-enzymatic antioxidants in the kidney compared to normal control rats, while α-Bisabolol treatment showed a significant (*p* < 0.05) decrease in the activities/levels of enzymatic and non-enzymatic antioxidants in the kidney compared to DOX-alone treated rats. Each column is mean ± SEM for eight rats in each group; columns not sharing a common symbol (*, **) differ significantly with each other (* *p* < 0.05 vs. normal control, ** *p* < 0.05 vs. DOX control).

**Figure 2 ijms-23-10528-f002:**
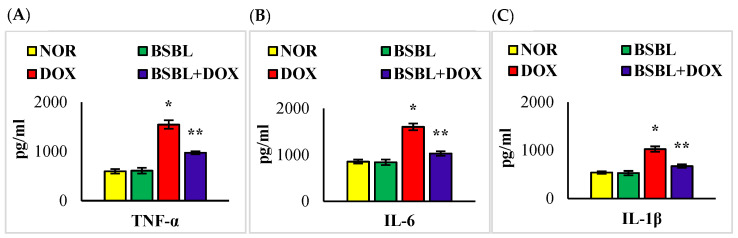
(**A**–**C**) Effect of α-Bisabolol on the levels of proinflammatory cytokines (TNF-α, IL-6, and IL-1β) in the kidney of normal and DOX-injected rats. DOX-injected rats showed a considerable (*p* < 0.05) increase in the levels of proinflammatory cytokines in the kidney compared to normal control rats, while α-Bisabolol treatment showed a significant (*p* < 0.05) decrease in the levels of proinflammatory cytokines in the kidney compared to DOX-alone treated rats. Each column is mean ± SEM for eight rats in each group; columns not sharing a common symbol (*, **) differ significantly with each other (* *p* < 0.05 vs. normal control, ** *p* < 0.05 vs. DOX control).

**Figure 3 ijms-23-10528-f003:**
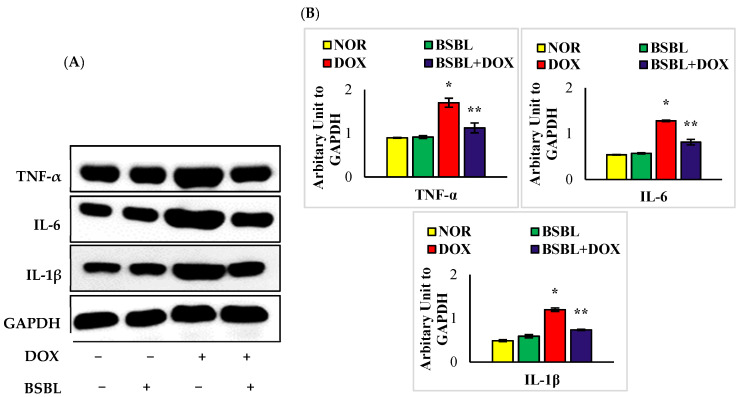
Effect of α-Bisabolol on the expressions of proinflammatory cytokines (TNF-α, IL-6, and IL-1β) in the kidney of normal and DOX-injected rats. (**A**). Representative images of Western immunoblot analysis for TNF-α, IL-6, and IL-1β. (**B**) Densitometric analysis of renal protein expressions of TNF-α, IL-6, and IL-1β assessed by Western blot analysis revealed that DOX-injected rats showed a considerable (*p* < 0.05) increase in the renal protein expressions of proinflammatory cytokines compared to normal control rats while α-Bisabolol treatment showed significant (*p* < 0.05) decrease in the expressions of cytokines in the kidney compared to DOX-alone treated rats. Immunoblotting analysis was done in duplicates; columns not sharing a common symbol (*, **) differ significantly with each other (* *p* < 0.05 vs. normal control, ** *p* < 0.05 vs. DOX control).

**Figure 4 ijms-23-10528-f004:**
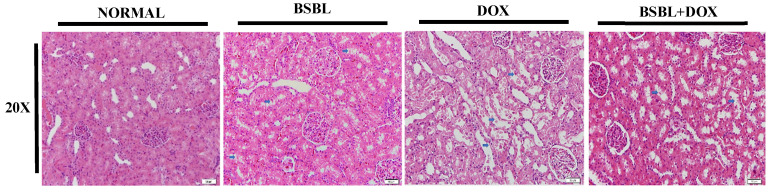
Histopathology of the kidney. Normal control and α-Bisabolol alone treated rat’s kidney showed no pathological changes. DOX-alone treated rats showed thinning and loss of apical cytoplasm, necrosis of individual tubular epithelial cells, loss of nuclei, adhesion, and denudation of tubular basement membrane, whereas α-Bisabolol treatment reinstates the near normal renal architecture in DOX-injected rats (Magnification; 20×).

**Figure 5 ijms-23-10528-f005:**
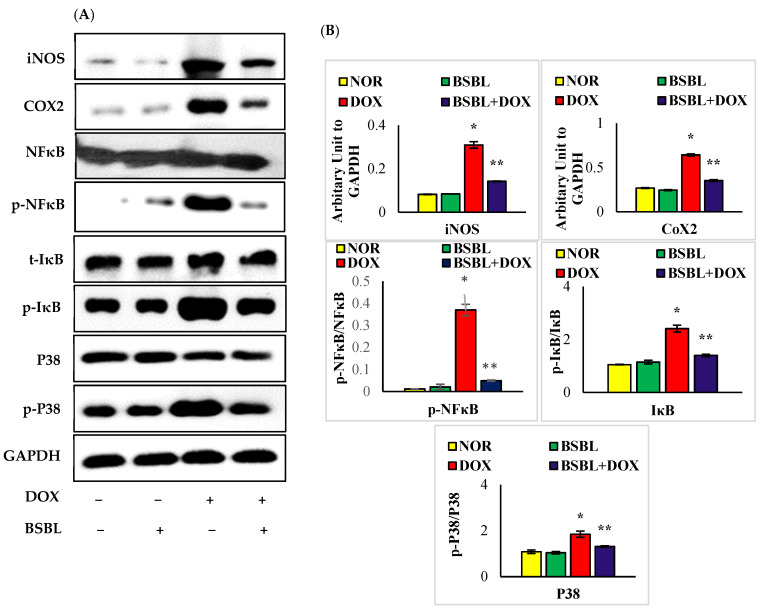
Effect of α-Bisabolol on NF-κB/MAPK signaling in the kidney of normal and DOX-injected rats. (**A**). Representative images of Western immunoblot analysis for iNOS, COX-2, p-NF-κB-P65, IκBα, p-IκBα, P38, and p-P38. (**B**) Densitometric analysis of renal protein expressions of iNOS, COX-2, p-NF-κB-P65, IκBα, p-IκBα, P38, and p-P38 assessed by Western blot analysis. The expressions of renal proinflammatory mediators (iNOS and COX-2) and NF-κB/MAPK signaling proteins in DOX-alone treated rats considerably (*p* < 0.05) increased compared to normal rats while α-Bisabolol treatment considerably (*p* < 0.05) reduced the altered renal expressions of inflammatory mediators (iNOS and COX-2) and NF-κB/MAPK signaling proteins compared to DOX-alone treated rats. Immunoblotting analysis was done in duplicates. Columns not sharing a common symbol (*, **) differ significantly with each other (* *p* < 0.05 vs. normal control, ** *p* < 0.05 vs. DOX control).

**Figure 6 ijms-23-10528-f006:**
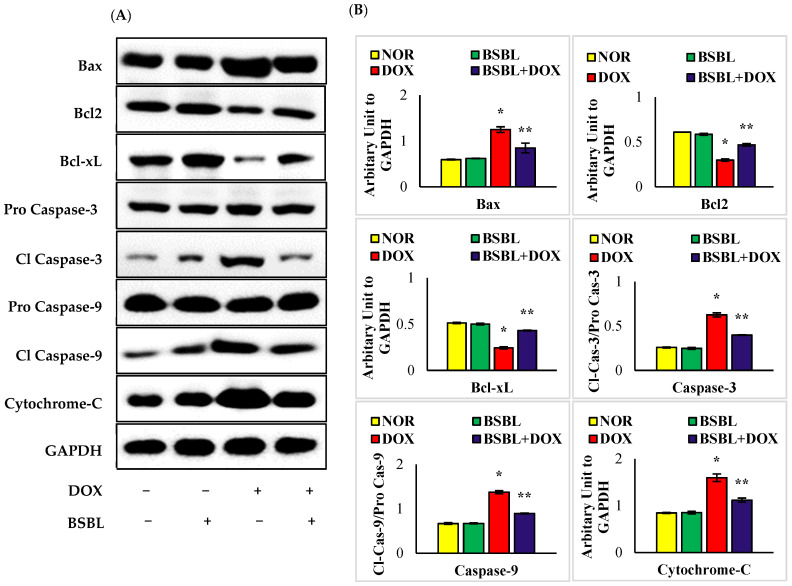
Effect of α-Bisabolol on the intrinsic apoptosis pathway in the kidney of normal and DOX-injected rats. (**A**). Representative images of Western immunoblot analysis for Bax, Bcl2, Bcl-xL, procaspase-3, active caspase-3, procaspase-9, active caspase-9, cytochrome-C. Immunoblotting analysis was done in duplicates. (**B**). Densitometric analysis of renal protein expressions of Bax, Bcl2, Bcl-xL, active caspase-3, active caspase-9, cytochrome-C. DOX-injected rats showed significant (*p* < 0.05) increase in the renal protein expressions of pro-apoptotic proteins (Bax, cleaved caspase-3, cleaved caspase-9 and cytochrome-C) with considerable (*p* < 0.05) decrease in the expressions of anti-apoptotic proteins (Bcl2 and Bcl-xL) compared to normal control rats. Meanwhile, α-Bisabolol treatment to DOX-injected rats significantly (*p* < 0.05) downregulates the renal protein expressions of Bax, cleaved caspase-3, cleaved caspase-9, and cytochrome-C with significant (*p* < 0.05) increase in the expression of renal Bcl2 and Bcl-xL compared to DOX-alone injected rats. Columns not sharing a common symbol (*, **) differ significantly with each other (* *p* < 0.05 vs. normal control, ** *p* < 0.05 vs. DOX control).

## Data Availability

This is a research article, and the majority of the articles referred to are cited appropriately in the manuscript.

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
