# Peer review of "α-Bisabolol Attenuates Doxorubicin Induced Renal Toxicity by Modulating NF-κB/MAPK Signaling and Caspase-Dependent Apoptosis in Rats"

_ijms, 2022, doi:10.3390/ijms231810528_

Round 1

Reviewer 1 Report

The manuscript entitled "α-Bisabolol attenuates doxorubicin induced renal toxicity by modulating NF-κB/MAPKsignaling and caspase-dependent apoptosis in rats" by Arunachalam et al, presented a study about using α-Bisabolol to ameliorate nephrotoxicity induced by doxorubicin in rats. Overall, this manuscript is clearly written. However, the major concern is whether the administration of α-Bisabolol will influence the therapeutic effects of doxorubicin. If the authors can address this concern, I might recommend its publication. Meanwhile, the authors should correct/revise the comments below

  1. Please elaborate on each Figure legend 
  2. The resolutions of Figure 3, 4, 5 are relatively low, please improve them.
  3. Please add the scale bars in Figure 4
  4. Pilcrow marks are found in Figure 3A and 5A, please remove them.
  5. In Figure 5A, are there any blots about phospho-NF-κB?

Author Response

Response to the Reviewer 1

We sincerely thank you for your comments to improve the manuscript. As advised, all the changes have been incorporated into the revised manuscript.

We have revised appropriately as desired for all the issues. We do hope that the revised manuscript will be up to the satisfaction of the respected reviewer.

  1. Please elaborate on each Figure legend.

Reply: Thanks for the suggestion. We have rewritten the figure legends in the revised manuscript.

  1. The resolutions of Figures 3, 4, 5 are relatively low, please improve them.

Reply: We changed the resolutions of all figures in the revised manuscript.

  1. Please add the scale bars in Figure 4.

Reply: We included scale bars for Figure 4.

  1. Pilcrow marks are found in Figures 3A and 5A, please remove them.

Reply: We removed the pilcrow marks in Figures 3A and 5A.

  1. In Figure 5A, are there any blots about phospho-NF-κB?

Reply: We included the blot for phospho-NF-κB in the revised manuscript.

Reviewer 2 Report

In the paper α-Bisabolol attenuates doxorubicin induced renal toxicity by modulating NF-κB/MAPK signaling and caspase-dependent apoptosis in rats the authors have tested the effect of α-Bisabolol in nephrotoxic rat model. They have found that treatment with α- Bisabolol prevented nephrotoxicity caused by Doxorubicin (DOX), restored antioxidants, diminished lipid peroxidation, prevented rise in proinflammatory cytokines, and favorably modulated the NF-κB/MAPK signaling pathway and apoptosis in kidneys of DOX-treated rats.

Comments 

11) Major characteristics of α-Bisabolol should be mentioned in the Abstract. In the text it should be added more about chemistry of α-Bisabolol. It is known that α-Bisabolol is used in cosmetics and as food additive and that is found to be beneficial in some diseases that include cancers. Is it known how much of bisabolol is reabsorbed to the blood from gastrointestinal tract and how much of it should be taken by food in order to exert beneficial effect in cancers and other diseases?  As bisabolol is natural monocyclic sesquiterpene alcohol and primary constituent of the essential oil from German chamomile (Matricaria recutita), Myoporum crassifolium, Salvia runcinata etc., could any other but dermal and oral route of its application be considered?

22) In the Figures, there is no need of repeating y-axis legend on the x-axis.  

33) Have α-Bisabolol only group received the substance for all 5 days of the experiment?

44) Figure 4 (histology) is over stained and tissue preparation and/or magnification do not allow us to see kidney structure, tubules etc. Only glomeruli are recognizable and limited tubular structure. Do authors have better kidney staining and could they use higher magnification (100X)? They could exclude 10X magnification images, as they don’t show the structure well. 

55) The authors have tested many pro-inflammatory and pro-apoptotic and anti-apoptotic molecules by Western blot. They actually measured p38, not p44/42 (Erk1/2), which is considered to be main MAPK. From the Western blots they have done, the authors could just speculate, based on expression of p38 and its phosphorylated form and findings in literature that activated p38 will be involved in NFkB stimulation. There are no experiments in this manuscript showing directly that giving p38 inhibitors in their particular model will prevent NFkB activation and have at least partial beneficial effect. These limitations should be mentioned in the paper.

66)  Because whole kidney extracts were used for Western blot it is not clear if there is difference in α-Bisabolol action between glomerular and tubular part of the kidneys. As histology is poor, it is not possible to get any clue about main α-Bisabolol action in injured kidneys.

77)  English should be checked and corrected; there are some repetitions and incorrectness in the text.

Author Response

Response to the Reviewer 2

We sincerely thank you for your comments to improve the manuscript. As advised, all the changes have been incorporated into the revised manuscript.

We have revised appropriately as desired for all the issues. We do hope that the revised manuscript will be up to the satisfaction of the respected reviewer.

 1) Major characteristics of α-Bisabolol should be mentioned in the Abstract. In the text, it should be added more about the chemistry of α-Bisabolol. It is known that α-Bisabolol is used in cosmetics and as a food additive and that is found to be beneficial in some diseases that include cancers. Is it known how much bisabolol is reabsorbed into the blood from the gastrointestinal tract and how much of it should be taken by food in order to exert beneficial effects on cancers and other diseases?  As bisabolol is natural monocyclic sesquiterpene alcohol and a primary constituent of the essential oil from German chamomile (Matricaria recutita), Myoporum crassifolium, Salvia runcinata etc., could any other but the dermal and oral route of its application be considered?

Reply: Thanks for your suggestion. We have included all the suggested details about α-bisabolol in the abstract and in the manuscript.

2) In the Figures, there is no need of repeating Y-axis legend on the x-axis. 

Reply: We have removed the x-axis legend in all figures. 

3) Have α-Bisabolol only group received the substance for all 5 days of the experiment?

Reply: Yes, animals in the α-bisabolol alone group received α-bisabolol (25 mg/kg, intraperitoneally) daily for a period of 5 days. We included this detail in the Experimental Design section of the revised manuscript.

4) Figure 4 (histology) is over-stained and tissue preparation and/or magnification do not allow us to see kidney structure, tubules, etc. Only glomeruli are recognizable and limited tubular structure. Do authors have better kidney staining and could they use higher magnification (100X)? They could exclude 10X magnification images, as they don’t show the structure well. 

Reply: Thanks for the suggestion. We replaced all the histological figures as per your suggestion in the revised manuscript. The histopathologist has well evaluated the observations.

5) The authors have tested many pro-inflammatory and pro-apoptotic and anti-apoptotic molecules by Western blot. They measured p38, not p44/42 (Erk1/2), which is the main MAPK. From the Western blots they have done, the authors could just speculate, based on the expression of p38 and its phosphorylated form and findings in the literature that activated p38 will be involved in NFkB stimulation. There are no experiments in this manuscript showing directly that giving p38 inhibitors in their model will prevent NF-kB activation and have at least a partial beneficial effect. These limitations should be mentioned in the paper.

Reply: Many thanks for this valuable suggestion. We have included the limitations of this study in the revised manuscript.

6)  Because whole kidney extracts were used for Western blot it is not clear if there is a difference in α-Bisabolol action between the glomerular and tubular parts of the kidneys. As histology is poor, it is not possible to get any clue about the main α-Bisabolol action in injured kidneys.

Reply: Now, we have replaced the histology figures as per your valuable suggestion with better resolution. The pathological changes in the kidneys are clearly visible in the replaced figures.

7)  English should be checked and corrected; there are some repetitions and incorrectness in the text.

Reply: Many thanks for your comments. Language editing was done. We removed the repetitions and corrected the common spelling and grammatical errors in the revised manuscript.

Round 2

Reviewer 1 Report

Thank you for addressing the comments and revising the manuscript. I would like to recommend its publication after authors' proofreading to correct the comments below:

1) In the Methods section, please provide the city and state of Cell signaling technology.

2) Please provide the company and its location details of the secondary antibodies in the Methods section. 

3) In figure legends, please provide the findings/result from the experiment instead of only a sentence of what experiment had been taken.

4) Please keep the NF-κ B in the blots of Figure 5. 

5) Some plots are overlapped or chopped in the manuscript, see Figure 1, 3, 5, and 6. Please make sure the full plots are provided.

Author Response

Reviewer-1

1) In the Methods section, please provide the city and state of Cell signaling technology.

Reply:  We included the city and state of Cell signaling technology in the methods section

2) Please provide the company and its location details of the secondary antibodies in the Methods section.

Reply:  We included the company and its location details of the secondary antibodies in the Methods section.

3) In figure legends, please provide the findings/result from the experiment instead of only a sentence of what experiment had been taken.

Reply:   As per your valuable suggestion, we elaborated the figure legends in the revised manuscript.

4) Please keep the NF-κ B in the blots of Figure 5. 

Reply:  We included the blot for NF-κ B in Figure 5.

5) Some plots are overlapped or chopped in the manuscript, see Figure 1, 3, 5, and 6. Please make sure the full plots are provided.

Reply:  We have corrected the Figures 1, 3, 5, and 6 in the revised manuscript.

Reviewer 2 Report

The authors have improved the paper considerably, especially kidney histology and images. The authors should add that human data from future studies are necessary to confirm their findings in animal model of renal toxicity induced by doxorubicin. 

Author Response

Reviewer-2

The authors have improved the paper considerably, especially kidney histology and images. The authors should add that human data from future studies are necessary to confirm their findings in animal model of renal toxicity induced by doxorubicin. 

Reply: We have included the suggested details in the revised manuscript.
